# The Prospective COVID-19 Post-Immunization Serological Cohort in Munich (KoCo-Impf): Risk Factors and Determinants of Immune Response in Healthcare Workers

**DOI:** 10.3390/v15071574

**Published:** 2023-07-18

**Authors:** Christina Reinkemeyer, Yeganeh Khazaei, Maximilian Weigert, Marlene Hannes, Ronan Le Gleut, Michael Plank, Simon Winter, Ivan Noreña, Theresa Meier, Lisa Xu, Raquel Rubio-Acero, Simon Wiegrebe, Thu Giang Le Thi, Christiane Fuchs, Katja Radon, Ivana Paunovic, Christian Janke, Andreas Wieser, Helmut Küchenhoff, Michael Hoelscher, Noemi Castelletti

**Affiliations:** 1Division of Infectious Diseases and Tropical Medicine, LMU University Hospital, LMU Munich, 80802 Munich, Germany; christina.reinkemeyer@med.uni-muenchen.de (C.R.); marlene.hannes@med.uni-muenchen.de (M.H.); michael.plank@med.uni-muenchen.de (M.P.); simon.winter@med.uni-muenchen.de (S.W.); ivan.norena@lrz.uni-muenchen.de (I.N.); raquel.rubio@med.uni-muenchen.de (R.R.-A.); ivana.paunovic@med.uni-muenchen.de (I.P.); christian.janke@lrz.uni-muenchen.de (C.J.); wieser@mvp.lmu.de (A.W.); hoelscher@lrz.uni-muenchen.de (M.H.); 2Statistical Consulting Unit StaBLab, Department of Statistics, LMU Munich, Ludwigstraße 33, 80539 Munich, Germany; yeganeh.khazaei@stat.uni-muenchen.de (Y.K.); maximilian.weigert@stat.uni-muenchen.de (M.W.); theresa.meier@stablab.stat.uni-muenchen.de (T.M.); lisa-xu@gmx.at (L.X.); simon.wiegrebe@stat.uni-muenchen.de (S.W.); kuechenhoff@stat.uni-muenchen.de (H.K.); 3Munich Center for Machine Learning (MCML), 80539 Munich, Germany; 4Institute of Computational Biology, Helmholtz Munich, 85764 Neuherberg, Germany; ronan.legleut@helmholtz-munich.de (R.L.G.); christiane.fuchs@helmholtz-munich.de (C.F.); 5Core Facility Statistical Consulting, Helmholtz Munich, 85764 Neuherberg, Germany; 6Department of Genetic Epidemiology, University of Regensburg, 93053 Regensburg, Germany; 7Department of Pediatrics, Dr. von Hauner Children’s Hospital, University Hospital, LMU Munich, Lindwurmstrasse 4, 80337 Munich, Germany; tlethi@med.lmu.de; 8Faculty of Business Administration and Economics, Bielefeld University, 33615 Bielefeld, Germany; 9Center for Mathematics, Technische Universität München, 85748 Garching, Germany; 10Institute and Outpatient Clinic for Occupational, Social and Environmental Medicine, University Hospital, LMU Munich, 80336 Munich, Germany; katja.radon@med.uni-muenchen.de; 11Center for International Health (CIH), University Hospital, LMU Munich, 80336 Munich, Germany; 12Comprehensive Pneumology Center (CPC) Munich, German Center for Lung Research (DZL), 89337 Munich, Germany; 13German Center for Infection Research (DZIF), Partner Site Munich, 80802 Munich, Germany; 14Fraunhofer Institute for Translational Medicine and Pharmacology ITMP, Immunology, Infection and Pandemic Research, 80799 Munich, Germany; 15Max von Pettenkofer Institute, Faculty of Medicine, LMU Munich, 80336 Munich, Germany; 16Institute of Radiation Medicine, Helmholtz Zentrum München, 85764 Neuherberg, Germany

**Keywords:** COVID-19, SARS-CoV-2, health care workers, vaccination, immunologic response, antibodies, seroprevalence, breakthrough infections, ORCHESTRA

## Abstract

Antibody studies analyze immune responses to SARS-CoV-2 vaccination and infection, which is crucial for selecting vaccination strategies. In the KoCo-Impf study, conducted between 16 June and 16 December 2021, 6088 participants aged 18 and above from Munich were recruited to monitor antibodies, particularly in healthcare workers (HCWs) at higher risk of infection. Roche Elecsys^®^ Anti-SARS-CoV-2 assays on dried blood spots were used to detect prior infections (anti-Nucleocapsid antibodies) and to indicate combinations of vaccinations/infections (anti-Spike antibodies). The anti-Spike seroprevalence was 94.7%, whereas, for anti-Nucleocapsid, it was only 6.9%. HCW status and contact with SARS-CoV-2-positive individuals were identified as infection risk factors, while vaccination and current smoking were associated with reduced risk. Older age correlated with higher anti-Nucleocapsid antibody levels, while vaccination and current smoking decreased the response. Vaccination alone or combined with infection led to higher anti-Spike antibody levels. Increasing time since the second vaccination, advancing age, and current smoking reduced the anti-Spike response. The cumulative number of cases in Munich affected the anti-Spike response over time but had no impact on anti-Nucleocapsid antibody development/seropositivity. Due to the significantly higher infection risk faced by HCWs and the limited number of significant risk factors, it is suggested that all HCWs require protection regardless of individual traits.

## 1. Introduction

The first report of the severe acute respiratory syndrome Coronavirus 2 (SARS-CoV-2) causing COVID-19 was on 31 December 2019 in the city of Wuhan (Hubei province, China) [1]. The World Health Organization (WHO) declared COVID-19 as a pandemic on 11 March 2020, after more than 118,000 cases in 114 countries and 4291 deaths occurred [2]. Since then, there have been outbreaks worldwide, with approximately 767 million confirmed cases and more than 6.9 million deaths as of June 2023 [3]. The first COVID-19 cases in Germany were observed in the municipality of Munich in late January 2020 [4]. Several vaccines were promptly developed and have been available in Germany since 27 December 2020 [5]. The first individuals to receive vaccinations were healthcare workers (HCW: people engaged in work actions whose primary intent is to improve health [6]) (HCWs), the elderly, and those who were at a high risk of severe illness to prevent the healthcare system from collapsing from overwhelming case numbers or lack of personnel [6,7,8,9]. HCWs are of particular interest and require careful investigation regarding SARS-CoV-2 infections. As vaccine protection diminishes over time, receiving an early vaccination reduces the risk of early infection but may increase the risk of later infection. This has been noted in several studies [9,10,11]. 

Many cohort studies have been set up since the beginning of the pandemic to analyze risk factors for infection before and after vaccination in both the general population [12,13,14,15,16] and HCWs [17,18]. 

Considering the role of antibody levels in protection against infection, most studies also analyze antibody titers over time. Anti-nucleocapsid (anti-N) antibodies develop only after natural infection (or vaccination with nucleocapsid-containing vaccines not commonly used in the Western world), while anti-spike (anti-S) antibodies develop after natural infection or/and vaccination [19].

Collatuzzo et al. [17] analyzed the predictors for a longer duration of the anti-S immune response at 9 months after the first COVID-19 vaccination in a multicentric European cohort of HCWs. A part of these data was fed into their analysis following the European-wide Consortium ORCHESTRA (Connecting European Cohorts to Increase Common and Effective Response to SARS-CoV-2 Pandemic). Female gender, young age, a previous infection, two vaccine doses, and mRNA and heterologous vaccination were found to determine higher anti-S antibody levels.

Moncunill et al. [20] analyzed determinants of antibody responses to COVID-19 mRNA vaccines in a cohort of exposed and naïve HCWs. Comparing previously SARS-CoV-2 infected versus uninfected individuals, the first ones were found to have higher anti-S IgA, IgG, and IgM levels, independently of the brand of the vaccine. At the same time, non-infected individuals developed significantly higher antibodies, depending on the brand of the vaccine. Interestingly, despite the clear impact of SARS-CoV-2 exposure on vaccine response, time since infection did not have a major effect on antibody response. Moreover, age and sex were not significantly associated with anti-S IgG levels in multivariable models.

Notarte et al. [21,22] analyzed determinants of antibody responses after COVID-19 mRNA vaccines in different populations. Regardless of the vaccine brand used, older age, male sex, seronegative status prior to vaccination, and presence of major comorbidities were associated with lower antibody titers (total antibodies, IgG, and/or IgA), supporting the findings of Yang [23]. 

Other factors leading to lower anti-S antibody titers were smoking [20,24] and homologous vaccination schemes [25,26,27].

In April 2020, the prospective Munich COVID-19 cohort (KoCo19) began to better evaluate the true case numbers [12,28,29]. Latest results show that vaccination prevents infection: anti-N seroprevalence was greater in the non-vaccinated population compared to the vaccinated one. At the same time, anti-N seroconversion rates (incidence) among vaccinated subjects did not show any statistical difference compared to the non-vaccinated group. Breakthrough infections (BTIs) may thus contribute relevantly to community spread, also considering the fact that the vaccinated population is much larger than the non-vaccinated population. The sub-cohort with jobs having a high contact risk with COVID-19 cases (e.g., HCWs) was found to have an increased risk for infection [30].

In May 2021, a new longitudinal cohort named KoCo-Impf (Prospective COVID-19 post-immunization Serological Cohort in Munich—Determination of immune response in vaccinated subjects) was established at the Division of Infectious Diseases and Tropical Medicine, comprising mostly HCWs with high contact risk with the SARS-CoV-2 virus. The analysis presented here aimed to identify risk factors for infection among HCWs, factors that influence the immune response following infection or vaccination, and differences between HCWs and the general population. The analysis utilized multivariable logistic regression analysis to identify risk factors for infection based on qualitative anti-N antibody results. Additionally, multivariable generalized linear models (GLM) were employed to determine the factors that raise antibody titers following infection and/or vaccination, using quantitative anti-N and anti-S antibody values. 

The KoCo-Impf study was recruited concurrently with the third and fourth follow-ups of KoCo19 in Munich. This allowed for a comparison of the general population of Munich (KoCo19) with their HCWs. Although the crude rates for anti-N seroprevalence were similar, a direct comparison was challenging. However, it was confirmed in both the KoCo19 and the KoCo-Impf that HCWs had a higher risk of infection. Sex, age, household size, and intake of immune-suppressing drugs were not found to be significant risk factors for infection in either cohort, but being a current smoker was.

## 2. Materials and Methods

### 2.1. The KoCo-Impf Cohort: Cohort Design, Inclusion Criteria, and Setting

The objective of KoCo-Impf is to investigate the short-, medium- and long-term immune response to SARS-CoV-2 vaccination. This study is funded by the European Union’s Horizon 2020 research and innovation program, as part of ORCHESTRA (Connecting European Cohorts to increase common and Effective SARS-CoV-2 Response), and also by the Division of Infectious Diseases and Tropical Medicine’s own resources [31].

Between 16 June and 16 December 2021, a total of 6467 participants aged 18 years or older, who had received at least one COVID-19 vaccination, were recruited for this study from the Munich municipality and surrounding areas. The recruitment campaign was carried out through three different paths (Figure 1, top): 

Path 1: At the local vaccination center Riem, where individuals were approached with this study’s information after their vaccination,

Path 2: At hospitals and nursing homes in the Munich area, targeting particularly exposed or vulnerable individuals (HCWs), and

Path 3: Via brochures and on the website of the Division of Infectious Diseases and Tropical Medicine for the general population.

Participants with language barriers (insufficient knowledge of the German language) or inability to provide informed consent were excluded.

Recruitment strategy, acquisition of informed consent, capillary blood samples, and questionnaire data occurred in different ways depending on the recruitment pathway:

Path 1: Directly after vaccination,

Path 2: By study teams making appointments on specific days to visit the sites, catching participants in the building during their working time, and

Path 3: Posting advertisements on the webpage of the Division of Infectious Diseases and Tropical Medicine, Klinikum der Universität München; participants could make an appointment for a personal visit via a hotline.

After data cleaning, 6088 participants were included in the analysis (Figure 1, bottom). Capillary blood samples were taken from participants to determine their antibody status, and questionnaire data were collected to obtain information on participants’ characteristics. The recruitment of employees from the University Hospital of Munich (LMU) was conducted simultaneously with the RisCoin HCWs cohort study, which studies risk factors for COVID-19 vaccine failure among HCWs [32].

### 2.2. Specimen Collection and Laboratory Analyses

Teams of trained field workers collected capillary blood samples (also known as a dry blood spot or DBS) following proper infectious disease control and blood sampling procedures to conduct laboratory analysis. The process for analyzing a DBS is explained in detail [33]. Two types of assays were used: the Roche Elecsys^®^ Anti-SARS-CoV-2 assay anti-Spike (anti-S) test, referred to as Ro-RBD-Ig, and the Roche Elecsys^®^ Anti-SARS-CoV-2 anti-Nucleocapsid (anti-N) test, referred to as Ro-N-Ig. The Ro-RBD-Ig detects antibodies after infection and vaccination, while the Ro-N-Ig test is used to differentiate between antibodies resulting from infection (both anti-S and anti-N present) and those due to vaccination (only anti-S present). The Ro-N-Ig test determines if an individual has previously had an infection but cannot provide information on the infection date. The Ro-RBD-Ig test has a cut-off value of 0.115 for DBS-seropositivity, while the Ro-N-Ig test has a cut-off value of 0.105. For both assays, a cross-reaction with viral infections predating the COVID-19 era could be excluded. This was achieved by analyzing samples obtained from blood donors prior to the emergence of COVID-19 [34,35].

### 2.3. Questionnaire Data 

This study used questionnaires to gather information from participants about

recruitment (institutional subgroup; recruitment date);demographic (date/year of birth; sex; level of education; household size);health-related behavior (smoking status; pre-existing medical conditions; medication scheme (intake of immunosuppressive drugs; others));employment-related behavior (occupational status; working conditions);COVID-19-related health status (vaccination status such as the date and type of first, second, and third vaccination if applicable; infection status, only Polymerase chain reaction (PCR)-confirmed COVID-19-diagnosis; diagnosis period; diagnosis date, month, and year; diagnosis in relation to vaccination and immunization status; diagnosis date after first vaccination; diagnosis date after full immunization (Two doses of Comirnaty, Spikevax or Vaxzevria or one dose of Jcovden at the time of data collection); severity of SARS-CoV-2-infection; previous contact with SARS-CoV-2 infected person; testing frequency; symptoms suggestive for COVID-19).

In the course of this study, three different versions of the questionnaire were used: Questionnaire 1 was provided on paper and used at the beginning of this study. Questionnaire 2 (used after 15 October 2021) was also provided on paper and included questions about the possibility of a third COVID-19 vaccination, as well as additional information that had emerged as potentially relevant during the course of this study (e.g., educational attainment, occupation, the presence of pre-existing conditions, and the course of COVID-19 disease). Questionnaire 3 was completed online by LMU employee hospital participants and requested the same information as Questionnaires 1 and 2.

Participants in Path 1 received Questionnaire 1 on the day of recruitment and filled it out during the recruitment procedures. Participants in Paths 2 and 3 were given the option to fill out Questionnaires 1 and 2 beforehand and bring them to the recruitment session or to fill them out on the day of recruitment. Participants in Path 2 also received Questionnaire 3 on the day of recruitment and were asked to fill it out during the recruitment procedures or as soon as possible thereafter.

Paper-based questionnaires were digitized using the software FormPro (version 3.1, OCR System GmbH, Leipzig, Germany, 2021).

### 2.4. Variables Definition

The variables that were used for the analysis are described in Table 1 and were selected following medical relevance. While most of the variables were obtained directly from the questionnaire, some of them were derived from other variables. The latter includes the vaccination scheme, time since the second vaccination, the occurrence of BTIs, time since infection, and the combination of the vaccination scheme and former infection, which is referred to as “immunity” hereafter. The recruitment process for KoCo-Impf was unique as it took place at various institutions over a period of seven months during the pandemic. Since a positive anti-N antibody level indicates a past infection, which could have occurred a long time ago, it is essential to take the different waves of the pandemic into account and correct for the different times at risk. Therefore, the cumulative number of new COVID-19 cases from the beginning of the pandemic to each date of recruitment was added as a covariate based on a weekly rolling window. A time lag of two weeks was applied, as anti-N and anti-S antibodies often need two weeks to develop after infection. [36,37].

Unlike most studies, we defined a SARS-CoV-2 infection by looking at anti-N antibody positivity instead of just considering PCR-positive tests. This approach ensures that asymptomatic and previously undiagnosed infections are more likely to be detected. Infection and vaccination by those vaccines used in our cohorts can be differentiated by serology, detecting both anti-S and anti-N antibodies. This analysis neglects information on symptoms. This choice was made due to the fact that many infections resulted in being asymptomatic, and the severity of symptoms does not necessarily indicate a different change in the antibody response.

### 2.5. Statistical Analyses 

Before conducting statistical analysis, data were cleaned and secured. Categorical variables are presented as frequencies and percentages, while continuous variables are presented as mean values and standard deviations (SD). Mean values, SDs, and crude associations were calculated for all quantities and are presented in Table 2.

To evaluate the risk of infection (anti-N seropositivity) based on qualitative binary anti-N results, a multivariable logistic regression model was used. Odds ratios (OR), 95% confidence intervals (CI), and *p*-values were computed. For the quantitative analyses, only participants with positive anti-N/S antibody values were included since the negative region is just affected by noise measurement and has no biological meaning. Two multivariable generalized linear models (GLM) with gamma distribution were fitted, with exponentiated coefficients representing the expected multiplicative changes in anti-N/S antibodies, 95% CIs, and *p*-values as output. To stabilize the anti-N model, fitting values greater than 10 were set to 10 (5 participants).

The covariate representing the cumulative number of COVID-19 cases detected in Munich (log-transformed to address the skewed distribution) was incorporated into all three models. This adjustment considered the different durations of potential exposure during the recruitment period. The covariables used in the three models are listed in Table 1, color-coded by model affiliation, and selected based on medical relevance. The missingness in the covariables was corrected by multiple imputations with m = 5 iterations. The response variables were also used in the multiple imputation procedure to obtain unbiased regression coefficients [38]. The total variance of the coefficient estimates over the repeated analyses was computed using Rubin’s rules [39]. The model evaluation was performed using (i) the area under the receiver operating characteristic curve (AUC) value obtained from a ten-fold cross-validation for the qualitative analysis of binary anti-N and (ii) diagnostics plots for the quantitative analyses (Appendix A).

All statistical analyses and visualization were performed using the R software (version 4.1.1, R Development Core Team, 2021). The models were estimated using the R package mgcv [40], and the visualization was conducted using the package APCtools [41].

## 3. Results

### 3.1. Cohort Description

Of a total of 6467 participants who were recruited for this study, 379 had to be excluded because of

missing or incomplete antibody measurements (*n* = 13);missing or implausible self-reported year of birth (*n* = 303);participation in clinical vaccination trials or recruitment after 16 December 2021 *n* = 27);vaccination with brands not authorized in Germany (*n* = 13);missing or diverse information on sex (*n* = 8);implausible vaccination dates (*n* = 3)unknown vaccination scheme (*n* = 12).

The final dataset that was analyzed included 6088 participants who were enrolled in 16 different institutional subgroups. All of these participants had complete measurements of anti-S/anti-N antibodies and self-reported questionnaire data (as shown in Figure 1). In total, 6088 participants were included in the qualitative binary anti-N model, 424 participants in the quantitative anti-N model, and 5750 participants in the quantitative anti-S model.

A description of the final cohort can be found in Table 2. Participants were aged from 18 to 96 years, with a mean/median age of 41.8/41.0. Thereof, 72.0% (4379/6088) were female, and 28.0% (1709/6088) male. The majority of study participants were HCWs in hospitals (79.8%, 4860/6088) or of other HC institutions (9.1%, 557/6088), while 11.0% (671/6088) were non-HCWs but from the general population. A total of 94.8% (5676/6088) of the participants were anti-S positive, while only 6.9% (424/6088) were anti-N positive. When the analysis was limited to HCWs, 6.9% (374/5417) were found to be anti-N positive.

### 3.2. Risk Factor Analysis for Anti-N Seropositivity

To determine the risk factors for contracting SARS-CoV-2, the qualitative anti-N serology test was used in conjunction with different covariables in a multivariable logistic regression model. The variables were selected following medical relevance and are described in Table 1. The results are presented in both Figure 2, where they are displayed as ORs, and in Appendix A, where they are displayed as logarithms of the ORs.

The results indicate that compared to the general population, there is a statistically significant positive association between being an HCW employed in a hospital and an increased risk of contracting the virus (Barmherzige Brüder 46.8 [22.1, 99.1], LMU Klinikum 8.6 [4.2, 17.6], MK Bogenhausen 10.0 [4.4, 22.2], MK Harlaching 9.7 [3.9, 23.8], MK Neuperlach 5.2 [1.6, 16.6], MK Schwabing 5.8 [2.4, 14.1], MK Thalkirchner Straße 7.8 [2.1, 28.3], MS Rümannstraße (6.3 [1.0, 40.9] and Seefeld 10.4 [3.0, 35.8]). This was also the case for HCWs employed in institutions of long-term care (Eichenau 46.6 [12.9, 168.3], MS Heilig Geist 29.9 [10.9, 82.1] and Obersendling 15.4 [3.5, 67.9]) and for HCWs employed in the vaccination center Riem (11.4 [5.4, 24.2]). Interestingly, two centers did not show a statistically significant association between being an HCW and an increased risk of infection (Tropical Institute (3.8 [0.7, 20.2]) and Friedenheimer Brücke (5.8 [0.6, 50.5]). The vaccination scheme analysis revealed a strong negative association for individuals vaccinated with two (0.03 [0.01, 0.05]) or three (0.02 [0.008, 0.04]) doses compared to unvaccinated individuals. Compared to non-vaccinated participants (353 individuals), no significant effect for a vaccination with one dose (380 individuals) could be found (0.6 [0.3, 1.1]). Participants reporting a past known contact with SARS-CoV-2-positives demonstrated a strong positive association with anti-N antibody seropositivity (2.2 [1.7, 2.8]) compared to those having none or unwitting contact. Interestingly, compared to non-smokers, a strong negative association could be detected only for current smokers (0.5 [0.3, 0.7]) (former smokers not significant 0.8 [0.5, 1.1]). Age (1.0 [0.9, 1.0]), sex (male 1.0 [0.8, 1.3]), household size (2 people 0.8 [0.6, 1.0], 3 people 0.9 [0.6, 1.3], 4 people 0.9 [0.6, 1.3], 5 people or more 0.9 [0.5, 1.5], intake of immunosuppressive drugs (yes 0.7 [0.3, 1.4]) and having had contact with patients (yes 1.1 [0.8, 1.5]) were not statistically significant associated with anti-N seropositivity. The cumulative cases in the Munich municipality, indicating the development of the pandemic, were also shown to be non-significant (2.5 [0.8, 7.5]).

### 3.3. Determinants of Antibody Response after SARS-CoV-2 Infection

To identify the factors that influence antibody responses following infection with SARS-CoV-2, the quantitative anti-N serology was associated with different covariables in a multivariable GLM with gamma distribution. The variables were selected following medical relevance and are described in Table 1. The findings of this analysis are presented in Figure 3 as the expected multiplicative changes in anti-N/S antibodies (exponentiated coefficients) and in Appendix A as coefficients of the model. The vaccination scheme analysis revealed that individuals with two (0.4 [0.2, 0.9]) and three vaccination doses (0.3 [0.1, 0.9]) had lower anti-N antibody levels compared to unvaccinated ones. No significant effect was found for participants with one vaccination dose (0.6 [0.3, 1.2]). A negative association could be detected for current smokers (0.6 [0.4, 1.0]), compared to non-smokers (former smokers not significant 1.1 [0.7, 1.7]). Age as a continuous variable was found to be a significant determinant, with older participants demonstrating higher anti-N antibody levels compared to younger ones (1.0 [1.003, 1.02]). Sex (male 1.2 [0.9, 1.6]), intake of immunosuppressive drugs (yes 1.1 [0.4, 2.8]), time since infection (three to less than six months ago 1.9 [0.1, 36.2], six to twelve months ago 1.3 [0.5, 3.2], more than twelve months ago 0.9 [0.3, 2.5]), BTI (yes 0.9 [0.4, 1.9]) and cumulative cases (1.9 [0.5, 6.9]) were not significant.

### 3.4. Determinants of Antibody Response after SARS-CoV-2 Vaccination and/or Infection

To ascertain the determinants that impact the antibody response after SARS-CoV-2 vaccination and infection, the quantitative anti-S serology was associated with different covariables in a multivariable GLM with gamma distribution. The results are presented in Figure 4 as the expected multiplicative changes in anti-N/S antibodies (exponentiated coefficients) and Appendix A as coefficients of the model. Compared to unvaccinated but infected individuals, a strong positive association could be found for participants who were vaccinated one (4.4 [1.6, 12.2]), two (23.4 [8.4, 64.8]), or three (469.5 [162.9, 1352.8]) times but did not undergo an infection. An even stronger positive association was found for participants who were vaccinated one (15.9 [6.3, 40.0]) or two (51.0 [20.9, 124.8]) times and underwent an infection. The group that received three vaccinations in addition to a past infection had a lower estimate (81.9 [20.6, 325.0]) compared to the group with three vaccinations but no previous infection. However, the estimate was still higher than the group that had received two vaccinations and had a history of infection. Moreover, days since the second vaccination and thus completion of the primary vaccination schedule revealed a high negative association (0.994 [0.993, 0.995]). Participants with BTI (infection occurring two weeks after the second vaccination) demonstrated a positive association compared to non-BTI infections (infection prior to or within two weeks after the second vaccination) (4.0 [2.2, 7.4]). Interestingly, the cumulative cases in the Munich municipality, indicating the development of the pandemic, were also shown to be significant (2.5 [1.6, 3.8]). Age was found to be a significant determinant, with older participants demonstrating a negative association with anti-S antibody quantity compared to younger participants (0.987 [0.983, 0.992]). Compared to non-smokers, a negative association could be detected for current smokers (0.8 [0.6, 0.9]) (former smokers not significant 1.0 [0.8, 1.1]). Time since infection (three to less than six months ago 0.7 [0.2, 2.6], six to twelve months ago 1.5 [0.6, 3.8], more than twelve months ago 1.3 [0.5, 3.4], no infection 0.4 [0.1, 1.2]), as well as sex (male 0.9 [0.8, 1.0]) and intake of immunosuppressive drugs (yes 1.1 [0.8, 1.5]) were not statistically significantly associated with quantitative anti-S serology. 

## 4. Discussion

In this study, we explore the factors contributing to COVID-19 infections in a cohort comprising both the general population and HCWs, who face an increased risk of exposure to the SARS-CoV-2 virus. We utilized capillary blood samples to detect the presence of SARS-CoV-2 antibodies, which are indicative of previous infections, including both symptomatic and asymptomatic cases, as well as vaccination history. Moreover, our analysis aimed to identify factors that influence the immune response following infection or vaccination.

The recruitment process for KoCo-Impf took place over a period of seven months during the waves of the pandemic. To consider the changing time under risk, we included the overall cumulative number of cases in Munich at the respective recruitment time as a continuous covariate in our analysis. Our analysis showed that this variable has a positive though not significant, effect on anti-N seropositivity, indicating that HCWs were only weakly affected by the infection waves of the general population. One possible explanation is that since most of the reported infections occurred between six and twelve months prior to blood sampling, they mostly occurred in the first half of 2021. As a result, any association between the cumulative number of cases and anti-N seropositivity in the second half of 2021 may not be evident. Another reason could be that localized outbreaks within specific institutions strongly influence the observed differences. This could potentially overshadow the effects of broader waves occurring within the general population. Other reasons could be that there was increasing availability of personal protective equipment (PPE) [42] and changes in risk behavior in 2021 [43]. In Bavaria, wearing protective FFP2 masks became mandatory in January 2021. Additionally, restrictions on access to public life were introduced in August 2021, based on vaccination, infection, and testing status, to reduce transmission rates [43]. As PPE has been shown to reduce the risk of infection [44], the increasing use of PPE may have compensated for any emerging outbreaks in 2021. In contrast, we found that the cumulative cases had an impact on anti-S antibody response, which could be explained by the different immune solicitations during the different waves. The dominant virus variant in Germany changed from alpha to delta in June 2021 [45], and a heterologous vaccination scheme was recommended from July 2021 onward [25,26,27,46,47]. Vaccination with Comirnaty rather than Spikevax was recommended for individuals younger than 30 years in November 2021 [48].

Age was found to be a statistically significant factor in anti-N immune response, with older participants showing higher levels after infection compared to younger ones. This is consistent with previous research that found a correlation between higher levels of the anti-N antibody and older age, male gender, ethnicity, and prior symptom history [49,50,51]. This suggests that infections in elderly individuals could lead to a more severe course of the disease and higher production of antibodies. In contrast to the anti-N immune response, our study showed that older age results in a decreased anti-S immune response, which is consistent with previous studies [21,22,52,53]. This suggests that the stimulation caused by vaccinations is more effective in younger individuals when compared to older ones. 

Another aspect to consider when examining the pattern of higher anti-N levels after infection but generally lower anti-S levels in non-infected individuals of higher age is the longitudinal development of the immune response in relation to the time since vaccination. Since older individuals are considered a “high-risk” group, they were vaccinated earlier than younger individuals [6,7,8]. Considering that anti-S antibodies follow a pattern of rising, peaking, falling, and eventually reaching a plateau [53], the earlier timing of vaccination could have led to a decrease in the anti-S antibody titer at the time of blood collection, resulting in a lower overall level. Consequently, the protection against a second infection is considered to be lower in this group, posing an increased risk of SARS-CoV-2 infection and a stronger immune response against the N protein compared to younger individuals who were recently vaccinated and had a higher anti-S antibody titer shortly after vaccination.

However, it is worth noting that a systematic review and meta-analysis conducted by Cheng et al. (2022) focused on prime-boost immunization with the COVID-19 vaccine but only analyzed studies with non-infected participants [27]. Subgroup analyses by age did not find a significant difference in antibody concentrations between young and old populations. Nevertheless, this finding may be attributed to the selection bias of only analyzing non-infected individuals. Young and elderly people who were most affected by the pandemic were excluded, and the definition of non-infected might vary between studies (RT-PCR and serology).

Our analysis has shown that individuals who currently smoke have a lower prevalence of anti-N SARS-CoV-2 antibodies compared to those who never smoked. It is important to note that the current smoker group in our cohort had significantly fewer participants compared to the non-smoker group (1 to 4 ratio). This discrepancy in sample size raises concerns about the comparability of the two groups, as the underrepresentation of current smokers may introduce bias to the results. However, the lower risk of infection among current smokers aligns with similar findings from the analysis of the KoCo19 cohort [30]. Additionally, a recent study by Günther et al. (2022) supports these findings, as it demonstrated that current smokers were nearly half as likely to test positive for SARS-CoV-2 antibodies compared to non-smokers [54]. That study did not observe any differences in antibody levels between smokers and non-smokers who had been infected with or vaccinated against SARS-CoV-2, suggesting that the lower prevalence of antibodies in smokers may be attributed to lower infection rates rather than reduced antibody response. In contrast, our results show a significantly reduced response to both the anti-S and anti-N antibodies in current smokers compared to non-smokers, consistent with previous studies by Reusch (2023), Ferrara (2022), and Moncunill (2022) [20,24,52]. Smoking may induce an immunosuppressive effect, as reported by Haddad (2021) and Sopori (2002) [55,56]. The lower anti-N antibody levels in current smokers compared to never-smokers may indicate not only a reduced development of antibodies but also a faster seroconversion to negative levels. Therefore, the anti-N seropositivity in current smokers may not be directly comparable to the never-smoker group, assuming a similar decrease and subsequent non-detection of past cases. It is also worth considering that smoking has been identified and communicated through the media as a risk factor for severe COVID-19 infections, leading to increased morbidity and mortality. Hence, it cannot be excluded that current smokers may have taken more precautions to avoid contact compared to non-smokers. The effect of current smoking on the risk of infection remains controversial and should be interpreted with caution [57].

The risk factor analysis showed that HCWs had an increased risk of infection compared to the general population, which is interestingly consistent with previous research on the KoCo19 cohort and other studies that have identified HCWs as a vulnerable group for infection [30,44,54]. However, the use of PPE has been shown to reduce the risk of infection [44], possibly leading to a change in the risk of infection in HCWs over time. Since our definition of infection is based only on positive anti-N, which remains positive for a long period of time [58], this baseline analysis of our study is not designed to detect this aspect. Recent research by Vivaldi et al. (2022) identified a change in the risk of infection due to time and vaccination status, with HCWs being at a higher risk of infection before vaccination but a reduced risk of breakthrough infection after primary vaccination [14]. Since the inclusion criteria for the KoCo-Impf study required at least one vaccination, it is impossible to correct this effect here. However, a follow-up analysis with the KoCo19 and the KoCo-Impf cohort may provide more insight into this aspect.

Another approach to determining whether HCWs have an increased risk of SARS-CoV-2 infection than the general population is by comparing anti-N seropositivities. In November 2021, the KoCo19 cohort, which represents the general Munich population, conducted its fourth follow-up in parallel with the KoCo-Impf recruitment. To compare the anti-N seroprevalence of both cohorts, we focused on the estimates for vaccinated persons in the KoCo19 cohort. The seropositivity was estimated to be 11.8% (9.8–13.8%) [30]. When we restricted the KoCo-Impf analysis to only HCWs, we observed a seroprevalence of 6.9% (6.2–7.6%), which is considerably lower than the seroprevalence of the vaccinated KoCo19 participants at the same time point. However, it is important to note that while the KoCo19 cohort is population-based and representative of the Munich population after statistical weighting, the KoCo-Impf cohort can be considered a convenience sample since it was not randomly selected. Therefore, it might be very complicated to compare both seroprevalences. This further emphasizes the importance of representative study designs. As the risk factor analysis for both KoCo19 and KoCo-Impf indicated a statistically significant higher risk of infections among HCWs, the lower seroprevalence in KoCo-Impf could be attributed to variations in infection and vaccination timing compared to the general population. Due to their higher risk, it is possible that HCWs were infected more frequently during the period when the general population was receiving their first two vaccinations. As HCWs, they had better access to testing facilities, which allowed them to become aware of their infection and receive vaccinations later in accordance with vaccination policies. On the other hand, in the general population, it is likely that more individuals were unknowingly infected and still received vaccinations despite their recent infection. The relatively lower underreporting probability among HCWs likely resulted in fewer cases where individuals were vaccinated despite having been recently infected, leading to lower seroprevalence among HCWs.

The risk of SARS-CoV-2 anti-N seropositivity was found to be higher among all HCWs except for those working in two specific institutions: Friedenheimer Brücke and Tropical Institute. While HCWs at Friedenheimer Brücke and the Tropical Institute have regular patient interactions, their work environment differs from that of HCWs in hospitals and long-term care facilities. Friedenheimer Brücke specializes in prenatal diagnostics, while the Tropical Institute primarily focuses on travel counseling and vaccinations. As a result, both facilities have a smaller patient population, and if symptomatic, these patients can choose to stay at home, thereby reducing the risk of infection for the personnel. The analysis did not find a significant effect of patient contact on SARS-CoV-2 anti-N seropositivity, suggesting that the increased risk of infection may be due to occupational activities and the working environment. This is consistent with recent research identifying occupational activities (tracheal intubation) as a risk factor for HCWs [44]. In addition, differences in infection frequency and spread between institutions can lead to variations in seropositivity rates.

As the institutional subgroup was found to have the strongest effect as a covariate, a sensitivity analysis was performed to evaluate how it impacted the overall risk factor analysis (Appendix A). However, no remarkable difference was observed.

Upon studying the kinetics of the anti-S antibody response, we found that the level increases with the number of COVID-19 vaccinations but decreases after days since the second vaccination. These results are consistent with previously published studies [52]. When individuals with one or two doses of vaccination were additionally infected, our analysis showed that they presented significantly higher anti-S values compared to only vaccinated individuals. Interestingly, with three vaccinations, the effect was reversed. While other studies with one or two vaccinations have shown similar behavior, we could not find comparable studies in the literature on the analysis of three vaccinations [21,22,52,53]. The combination of three vaccinations and one infection suggests that either the infection occurred in the early phases of the pandemic or recently (an infection between the vaccination scheme can be excluded in the time before Omicron), but the effect might be smaller due to the passage of time or ongoing immune response. This can be confirmed by the similarities with the estimate of two vaccinations with or without infection. 

Our findings also indicate that the sequence of the triggers is important, with BTIs showing higher anti-S antibody titers but a non-significant tendency towards lower anti-N. This is in line with the other literature where the interpretation is that the immune system is solicited with vaccination (higher anti-S) so that a severe disease can be prevented (lower anti-N, since less reaction is needed) [59,60,61].

It is interesting to note that even though SARS-CoV-2 infection clearly affects the anti-S immune response, the duration since infection did not have a significant effect in any of our models. This finding is consistent with results that have already been published [20]. Numerous studies have demonstrated that the longitudinal development of both anti-N and anti-S antibodies follows a pattern of increasing, peaking, decreasing, and ultimately plateauing [53,62,63]. In our data, most of the reported infections occurred between six and twelve months prior to blood sampling, during which time most participants had already reached the plateau phase. Therefore, the lack of statistical significance is likely due to the fact that the only trajectory that can be fitted to these data is the plateau phase. 

The analysis presented in this study encompasses the time period starting from the onset of the pandemic until December 2021. Therefore, the conclusions derived from this analysis specifically pertain to SARS-CoV-2 infections caused by the wild-type to delta variants of concern. A follow-up was carried out in May 2022 to include the circulation of the omicron variant of concern. A follow-up manuscript will present the findings of this follow-up study and compare them with the results obtained from the initial analysis.

The number of individuals who tested positive for anti-N antibodies but were unvaccinated (40) is lower than the number of individuals who tested positive for anti-S antibodies (53), even though their antibody response can only be attributed to a natural infection. This difference of 13 samples is likely due to the recruitment process rather than the assays themselves. Our cohort recruitment includes individuals at various stages of infection (recently infected, infected long ago, etc.) and at different time points of vaccination. Consequently, it is possible that a recently infected individual may only exhibit one type of antibody since their development requires time. On the other hand, someone who was infected a long time ago may have already seroconverted back, although not completely for both antibodies. Furthermore, information regarding vaccination status relies on self-reported questionnaire data, which may be influenced by bias or incomplete responses. Therefore, the discrepancy in this small number of samples is likely a combination of these factors.

After more than a year since the onset of the pandemic, we established the KoCo-Impf cohort to examine antibody development following vaccination and infection. Considering the significant findings already observed with KoCo19, we primarily focused on recruiting HCWs who face a specific risk of infection due to their frequent contact with multiple individuals, some of whom may be infected. It is important to note that our study population represents a convenience sample consisting solely of non-randomly selected vaccinated individuals. This aspect makes it more challenging to compare our results directly with those of the general population. However, the unique combination of our definition of seropositivity (based on anti-N and anti-S values) and the large sample size with detailed vaccination information makes our cohort unique in the world. We also found that vaccination protects against infection, but elderly people tend to have weaker immune responses and present higher anti-N but lower anti-S values compared to younger participants. Interestingly, smokers had a decreased risk of infection and lower immune responses after both vaccination and infection. HCWs were found to have a higher risk of SARS-CoV-2 infection in both the KoCo19 and the KoCo-Impf studies. However, only a few risk factors, such as age, vaccination status, contact with SARS-CoV-2 positive cases, and smoking status, were found to be statistically significant. As a result, no specific subgroups of HCWs requiring greater protection were identified. Instead, it is crucial to ensure the protection of all HCWs regardless of individual characteristics. 

## 5. Conclusions

HCWs had a higher risk of SARS-CoV-2 infection in both the KoCo19 and KoCo-Impf studies. Multiple vaccinations and diverse vaccination schedules reduced infection risk while influencing the anti-N and anti-S immune response. Age impacted immune response, with older individuals exhibiting differences compared to younger ones. Interestingly, smokers had a lower infection risk, but their immune response weakened after vaccination and infection. The limited number of significant risk factors indicates that no specific HCW subgroups require heightened protection but that the protection of all HCWs remains crucial, regardless of individual characteristics.

## Figures and Tables

**Figure 1 viruses-15-01574-f001:**
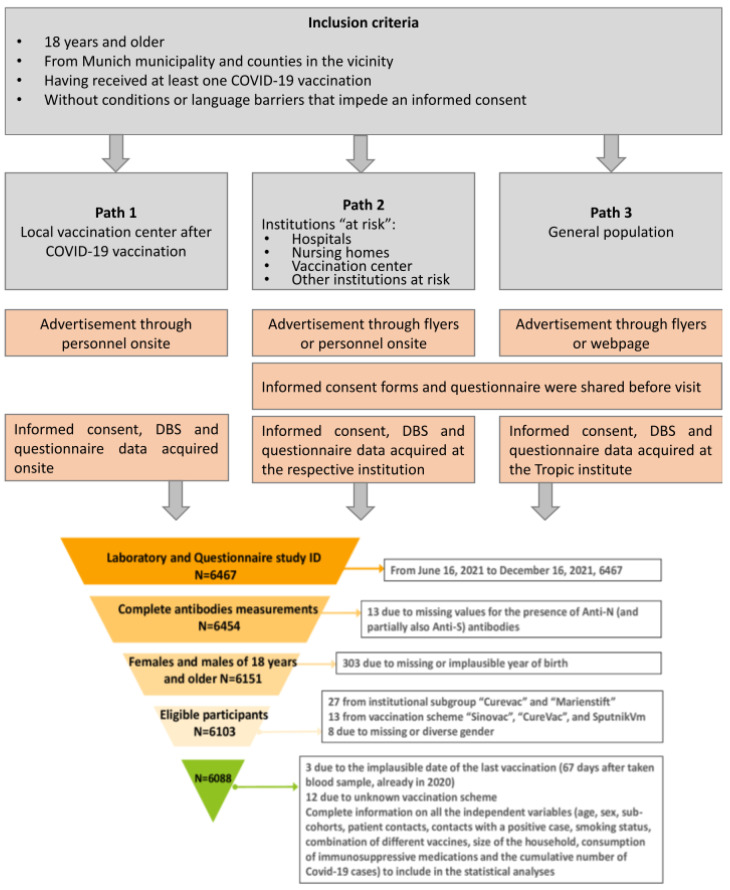
Recruitment paths and criteria for inclusion into the analysis. Gray boxes: inclusion criteria and places of recruitment. Orange boxes: information on advertisement modalities for recruiting participants; modalities of the acquisition of informed consent, questionnaire data, and capillary blood samples (acquired in person by study personnel). A triangle diagram describing the exclusion criteria and the final information of the analyzed participants.

**Figure 2 viruses-15-01574-f002:**
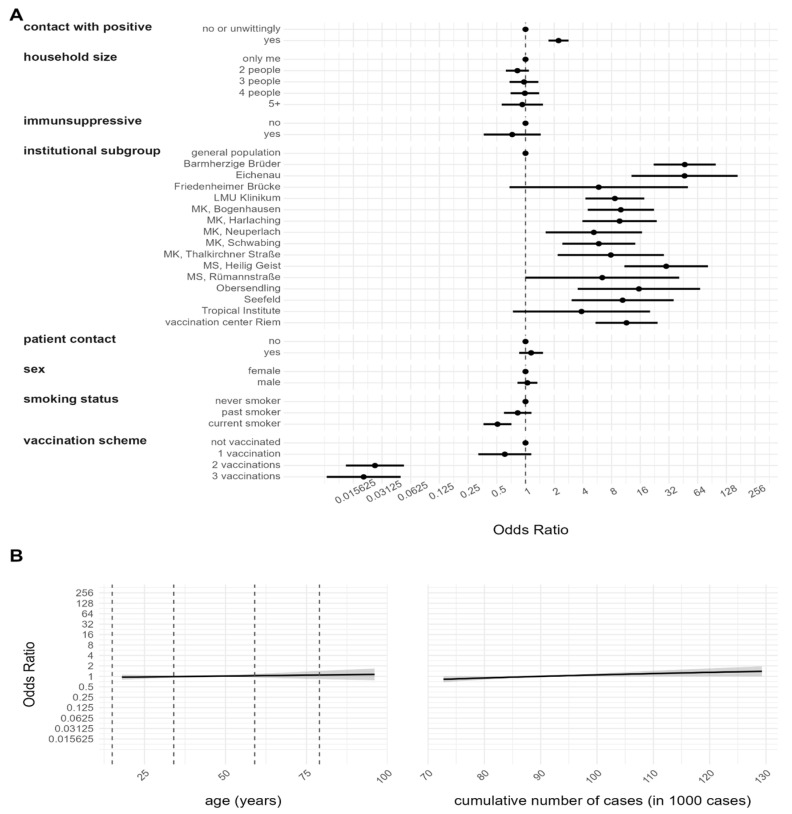
Risk factor analysis for SARS-CoV-2 infection, based on positive anti-N serology. Results are based on a logistic regression model and are given as ORs with a 95% CI. The obtained value of model evaluation using pooled AUC was 0.7398. (**A**) Estimates for categorical variables. (**B**) Estimates for continuous variables with 95% CI represented by the grey shadowed region.

**Figure 3 viruses-15-01574-f003:**
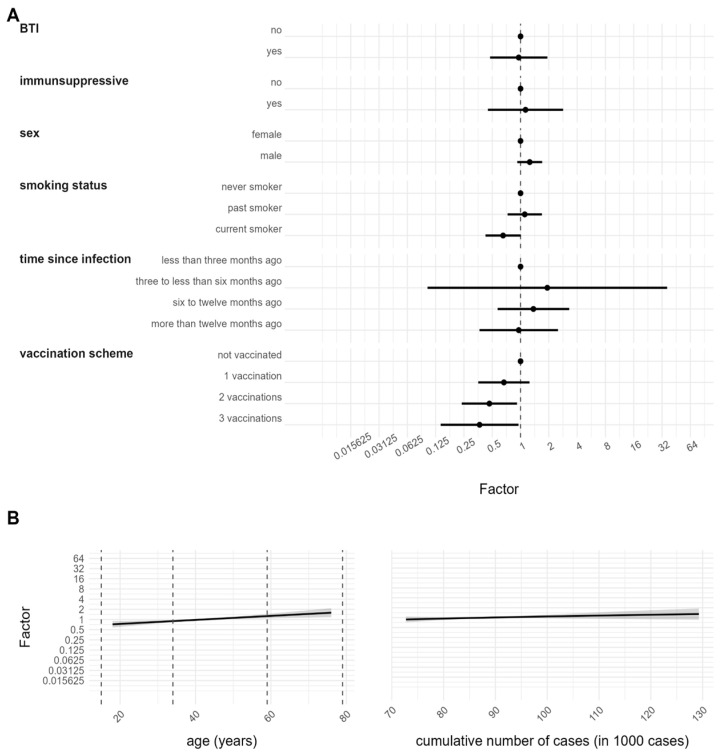
Anti-N antibody level after infection. Association between quantitative anti-N serology and determinants of antibody response. Results are based on a GLM with gamma distribution and are given as the expected multiplicative changes in anti-N/S antibodies (exponentiated coefficients) with a 95% CI.(**A**) Estimates for categorical variables. (**B**) Estimates for continuous variables with 95% CI represented by the grey shadowed region.

**Figure 4 viruses-15-01574-f004:**
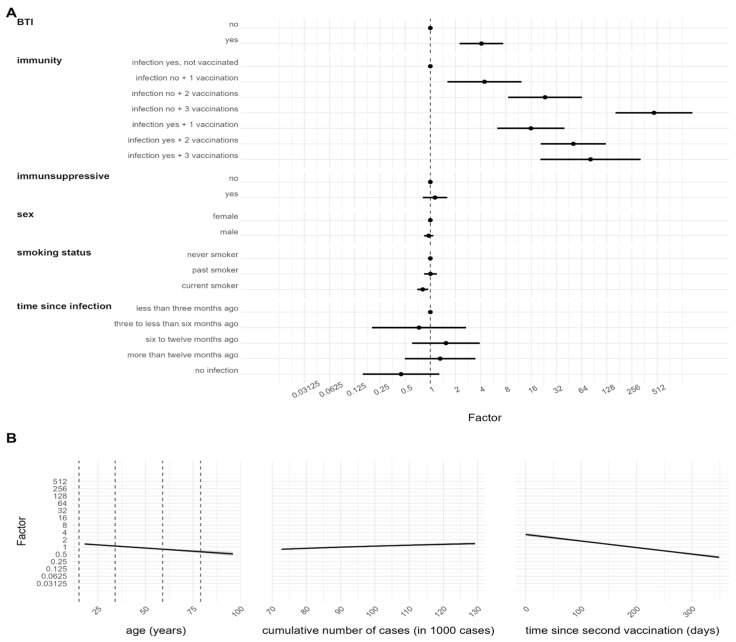
Anti-S antibody level after infection and vaccination. Association between quantitative anti-S serology and determinants of antibody response. Results are based on a GLM with gamma distribution and are given as the expected multiplicative changes in anti-N/S antibodies (exponentiated coefficients) with a 95% CI. (**A**) Estimates for categorical variables. (**B**) Estimates for continuous variables with 95% CI represented by the grey shadowed region.

**Table 1 viruses-15-01574-t001:** Variables description with color-coded allocation to the three statistical models used in the analysis. Covariables of: all three models, green; only anti-N qualitative model, pink; only anti-S quantitative model, gray; anti-N quantitative model and anti-S quantitative model, blue; anti-N qualitative model; anti-N quantitative model, gold.

Variable Name	Definition (Type of Variable)
Quantitative anti-N/S	The detected amount of Ro-N-Ig/Ro-RBD-Ig from DBS (continuous)
Qualitative anti-N/S	A positive anti-N/S result is defined when the amount of Ro-N-Ig/Ro-RBD-Ig is ≥0.105/0.115 (positive/negative)
Age ****	Age of participants in years (continuous)
Cumulative cases	Cumulative number of COVID-19 cases from the beginning of the pandemic till the recruitment date (continuous)
Intake of immunosuppressive drugs ****	Current intake of medications that may suppress the immune system (yes, no)
Sex ****	Sex of the participant (male, female)
Smoking status ****	Current smoking status (never smoker, current smoker, past smoker)
Contact with patients ****	Direct contact with patients (yes, no)
Contact with positives ****	Previous contact with COVID-19 affected/SARS-CoV-2 infected person (yes, no, or unwittingly)
Household size ****	Number of household members including participant (1, 2, 3, 4, 5, >5)
Institutional subgroup	Categorization according to the institution of recruitment (Hospitals *: Medical center of LMU, Tropical Institute **, MK Bogenhausen, MK Harlaching, MK Neuperlach, MK Schwabing, MK Thalkirchner Straße, Barmherzige Brüder, Seefeld, Institutions of long-term care: Eichenau, MS Heilig Geist, MS Rümannstraße, Obersendling Others: Vaccination center Riem, Friedenheimer Brücke, General population ***)
Breakthrough Infection (BTI) ****	An infection happened at least 2 weeks after the second dose (yes, no, not applicable)
Time since infection ****	Time between the sampling date and the positive PCR (infected in less than 3 months, infected between 3 and 6 months, infected between 6 and 12 months, infected after 12 months, no infection)
Combination of vaccination scheme and former infection (immunity)	A composite variable containing information on the previous infection (based on anti-N result) and the undergone vaccination scheme (infection yes, not vaccinated, infection yes + one vaccination, infection yes + two vaccinations, infection yes + three vaccinations, infection no + one vaccination, infection no + two vaccinations, infection no + three vaccinations)
Time since second vaccination ****	Time between the second vaccination and the sampling date (continuous)
Vaccination scheme ****	A combination of types of vaccination and number of vaccinations, including BioNTech/Pfizer, Moderna, AstraZeneca, Johnson & Johnson/Janssen (no vaccination, one vaccination, two vaccinations, three vaccinations)

* Includes study participants from Path 2. ** Division of Infectious Diseases and Tropical Medicine of LMU. *** Includes study participants from Path 1 and Path 3. **** Based on self-reported questionnaire data.

**Table 2 viruses-15-01574-t002:** Cohort description with data before imputation.

Covariate	Category	Number of Participants *N* (%)	Qualitative Anti-N *N* (%)		Qualitative Anti-S *N* (%)		Quantitative Anti-N Mean Value (SD)		Quantitative Anti-S Mean Value (SD)	
			Positive	Negative	Positive	Negative	Positive	Negative	Positive	Negative
Overall cohort		6088 (100.0)	424 (6.9)	5664 (93.1)	5767 (94.8)	321 (5.2)	0.94 (1.52)	0.06 (0.01)	83.54 (200.35)	0.03 (0.02)
Sex	Female	4379 (72.0)	296 (6.7)	4083 (93.3)	4199 (95.9)	180 (4.1)	0.88 (1.33)	0.06 (0.01)	82.39 (199.08)	0.03 (0.02)
	Male	1709 (28.0)	128 (7.4)	1581 (92.6)	1568 (91.8)	141 (8.2)	1.10 (1.86)	0.06 (0.01)	86.68 (204.17)	0.03 (0.02)
Institutional subgroup	Barmherzige Brüder	188 (3.0)	40 (21.2)	148 (78.8)	187 (99.5)	1 (0.5)	0.98 (1.04)	0.07 (0.008)	55.02 (106.23)	0.06 (NA)
	Eichenau	34 (0.5)	5 (14.7)	29 (85.3)	34 (100.0)	0 (0.0)	1.59 (2.00)	0.07 (0.004)	447.20 (427.47)	- *
	Friedenheimer Brücke	34 (0.5)	1 (2.9)	33 (97.1)	34 (100.0)	0 (0.0)	0.88 (NA)	0.08 (0.006)	82.45 (122.71)	-
	General population	671 (11.0)	50 (7.5)	621 (92.5)	366 (54.6)	306 (45.4)	1.33 (2.25)	0.07 (0.02)	43.84 (121.03)	0.03 (0.02)
	Medical Center of LMU	3689 (60.6)	213 (5.7)	3476 (94.3)	3680 (99.8)	9 (0.2)	0.86 (1.53)	0.06 (0.01)	85.62 (205.49)	0.04 (0.04)
	MK, Bogenhausen	238 (3.9)	23 (9.6)	215 (90.4)	238 (100.0)	0 (0.0)	1.42 (1.78)	0.07 (0.01)	62.67 (172.21)	-
	MK, Harlaching	154 (2.5)	14 (9.1)	140 (90.9)	154 (100.0)	0 (0.0)	0.87 (1.19)	0.07 (0.006)	43.20 (60.97)	-
	MK, Neuperlach	112 (1.8)	5 (4.4)	107 (95.6)	112 (100.0)	0 (0.0)	0.45 (0.38)	0.07 (0.005)	33.44 (32.95)	-
	MK, Schwabing	281 (4.6)	13 (4.7)	268 (95.3)	281 (100.0)	0 (0.0)	0.36 (0.35)	0.07 (0.009)	48.08 (128.11)	-
	MK, Thalkirchner Straße	67 (1.1)	4 (5.9)	63 (94.1)	67 (100.0)	0 (0.0)	2.15 (2.27)	0.07 (0.006)	40.60 (46.19)	-
	MS, Heilig Geist	60 (0.9)	14 (23.3)	46 (76.7)	60 (100.0)	0 (0.0)	0.61 (0.69)	0.06 (0.02)	140.81 (380.16)	-
	MS, Rümannstraße	36 (0.5)	2 (5.5)	34 (94.5)	36 (100.0)	0 (0.0)	0.58 (0.67)	0.06 (0.005)	531.93 (574.09)	-
	Obersendling	27 (0.4)	4 (14.8)	23 (85.2)	27 (100.0)	0 (0.0)	0.88 (0.66)	0.08 (0.004)	54.03 (113.73)	-
	Seefeld	83 (1.3)	5 (6.1)	78 (93.9)	83 (100.0)	0 (0.0)	1.26 (0.52)	0.06 (0.01)	138.71 (285.03)	-
	Tropical Institute	48 (0.8)	2 (4.1)	46 (95.9)	46 (95.9)	2 (4.1)	0.16 (0.05)	0.07 (0.01)	78.37 (115.27)	0.05 (0.02)
	Vaccination center Riem	366 (6.0)	29 (7.9)	337 (92.1)	363 (99.2)	3 (0.8)	0.76 (0.85)	0.07 (0.007)	101.04 (148.18)	0.06 (0.04)
Contact with patients	Yes	3505 (57.5)	261 (7.4)	3244 (92.6)	3493 (99.7)	12 (0.3)	0.90 (1.42)	0.06 (0.01)	94.39 (227.33)	0.03 (0.03)
	No	1833 (30.2)	111 (6.1)	1722 (93.9)	1647 (89.9)	186 (10.1)	0.89 (1.39)	0.06 (0.02)	65.44 (140.44)	0.03 (0.02)
	Unknown **	750 (12.3)	52 (6.8)	698 (93.2)	627 (83.8)	123 (16.2)	1.26 (2.09)	0.07 (0.02)	70.64 (167.82)	0.03 (0.02)
Contact with positives	Yes	2804 (45.9)	278 (9.9)	2526 (90.1)	2747 (97.9)	57 (2.1)	1.00 (1.62)	0.06 (0.01)	89.99 (215.54)	0.03 (0.02)
	No or unwittingly	3284 (54.1)	146 (4.4)	3138 (95.6)	3020 (91.9)	264 (8.1)	0.84 (1.28)	0.06 (0.01)	77.70 (185.37)	0.03 (0.02)
Smoking status	Never smoker	4177 (68.5)	315 (7.5)	3862 (92.5)	3967 (94.9)	210 (5.1)	0.96 (1.57)	0.06 (0.02)	86.29 (205.12)	0.03 (0.02)
	Current smoker	1062 (17.5)	49 (4.6)	1013 (95.4)	1009 (95.1)	53 (4.9)	0.52 (0.61)	0.06 (0.01)	73.95 (188.65)	0.03 (0.02)
	Past smoker	798 (13.1)	56 (7.1)	742 (92.9)	740 (92.8)	58 (7.2)	1.20 (1.71)	0.07 (0.01)	82.21 (190.29)	0.03 (0.02)
	Unknown	51 (0.9)	4 (7.8)	47 (92.2)	51 (100.0)	0 (0.0)	0.91 (0.65)	0.06 (0.007)	80.02 (201.80)	-
Vaccination scheme	No vacc. ***	353 (5.7)	40 (11.3)	313 (88.7)	53 (15.0)	300 (85.0)	1.65 (2.64)	0.07 (0.02)	13.25 (50.72)	0.03 (0.02)
	One vaccination	380 (6.1)	123 (32.5)	257 (67.5)	367 (96.6)	13 (3.4)	1.15 (1.53)	0.07 (0.01)	98.05 (226.56)	0.04 (0.04)
	Two vaccinations	5001 (82.2)	245 (4.9)	4756 (95.1)	4997 (99.9)	4 (0.1)	0.75 (1.23)	0.06 (0.01)	55.40 (136.23)	0.06 (0.03)
	Three vaccinations	354 (5.8)	16 (4.4)	338 (95.6)	350 (98.9)	4 (1.1)	0.79 (1.07)	0.06 (0.01)	480.65 (416.65)	0.04 (0.04)
Household size	One person	1586 (25.9)	117 (7.3)	1469 (92.7)	1477 (93.2)	109 (6.8)	1.01 (1.57)	0.06 (0.01)	80.86 (197.26)	0.03 (0.02)
	2 people	2219 (36.5)	140 (6.3)	2079 (93.7)	2107 (94.9)	112 (5.1)	1.08 (1.65)	0.06 (0.01)	84.91 (209.09)	0.03 (0.02)
	3 people	969 (15.8)	68 (7.1)	901 (92.9)	924 (95.4)	45 (4.6)	0.89 (1.53)	0.06 (0.01)	82.79 (172.72)	0.04 (0.03)
	4 people	890 (14.8)	67 (7.6)	823 (92.4)	859 (96.6)	31 (3.4)	0.70 (1.13)	0.06 (0.01)	83.94 (213.37)	0.02 (0.02)
	5 people or more	331 (5.4)	23 (6.9)	308 (93.1)	314 (94.9)	17 (5.1)	0.50 (0.67)	0.06 (0.01)	92.08 (205.55)	0.04 (0.03)
	Unknown	93 (1.5)	9 (8.8)	84 (91.2)	86 (93.2)	7 (6.8)	1.15 (2.29)	0.07 (0.01)	68.55 (163.15)	0.04 (0.03)
Intake of immunosuppressive drugs	Yes	178 (2.9)	11 (6.1)	167 (93.9)	166 (93.3)	12 (6.7)	1.09 (1.21)	0.06 (0.02)	103.35 (234.73)	0.03 (0.02)
	No	5855 (96.0)	406 (6.9)	5449 (93.1)	5550 (94.8)	305 (5.2)	0.94 (1.53)	0.06 (0.01)	82.39 (199.94)	0.03 (0.02)
	Unknown	55 (1.1)	7 (10.9)	48 (89.1)	51 (93.8)	4 (6.2)	0.81 (0.64)	0.06 (0.008)	144.25 (233.67)	0.01 (0.01)
Time since infection	Less than three months ago	11 (0.1)	7 (63.6)	4 (36.4)	10 (90.9)	1 (9.1)	0.74 (1.53)	0.03 (0.03)	835.43 (653.70)	0.04 (NA)
	Three to less than six months ago	10 (0.1)	3 (30.0)	7 (70.0)	10 (100.0)	0 (0.0)	0.74 (1.00)	0.05 (0.03)	184.22 (387.26)	-
	Six to twelve months ago	81 (1.3)	57 (70.3)	24 (29.7)	81 (100.0)	0 (0.0)	1.04 (1.75)	0.06 (0.03)	357.00 (500.08)	-
	More than twelve months ago	118 (1.9)	71 (59.6)	47 (40.4)	116 (98.4)	2 (1.6)	0.76 (1.10)	0.06 (0.02)	221.56 (301.96)	0.05 (0.05)
	No infection	5582 (91.8)	0 (0.0)	5582 (100.0)	5268 (94.4)	314 (5.6)	-	0.06 (0.01)	67.39 (166.41)	0.03 (0.02)
	Unknown	286 (4.8)	286 (100.0)	0 (0.0)	282 (98.7)	4 (1.3)	0.98 (1.56)	-	220.78 (323.30)	0.05 (0.03)
Breakthrough Infection (BTI)	Yes	63 (1.1)	28 (46.4)	35 (53.6)	62 (98.6)	1 (1.4)	0.58 (0.85)	0.05 (0.03)	546.24 (532.41)	0.09 (NA)
	No	6018 (98.8)	396 (6.5)	5622 (93.5)	5698 (94.8)	320 (5.2)	0.97 (1.55)	0.06 (0.01)	78.58 (187.13)	0.03 (0.02)
	Not applicable	7 (0.1)	0 (0.0)	7 (100.0)	7 (100.0)	0 (0.0)	-	0.07 (0.02)	21.87 (19.57)	-
Vaccination scheme and infection (immunity)	Infection yes, not vaccinated	40 (0.7)	40 (100.0)	0 (0.0)	36 (90.0)	4 (10.0)	1.65 (2.64)	-	18.30 (60.95)	0.05 (0.03)
	Infection yes + one vaccination	123 (2.0)	123 (100.0)	0 (0.0)	123 (100.0)	0 (0.0)	1.15 (1.53)	-	238.20 (341.43)	-
	Infection yes + two vaccinations	245 (4.0)	245 (100.0)	0 (0.0)	245 (100.0)	0 (0.0)	0.75 (1.23)	-	294.99 (398.29)	-
	Infection yes + three vaccinations	16 (0.3)	16 (100.0)	0 (0.0)	16 (100.0)	0 (0.0)	0.79 (1.07)	-	437.20 (462.30)	-
	Infection no, not vaccinated	313 (5.1)	0 (0.0)	313 (100.0)	17 (5.5)	296 (94.5)	-	0.06 (0.02)	2.56 (7.37)	0.03 (0.02)
	Infection no + one vaccination	257 (4.1)	0 (0.0)	257 (100.0)	244 (94.9)	13 (5.1)	-	0.07 (0.01)	27.40 (62.10)	0.04 (0.03)
	Infection no + two vaccinations	4756 (78.3)	0 (0.0)	4756 (100.0)	4752 (99.9)	4 (0.1)	-	0.06 (0.01)	43.06 (90.88)	0.06 (0.02)
	Infection no + three vaccinations	338 (5.5)	0 (0.0)	338 (100.0)	334 (98.9)	4 (1.1)	-	0.06 (0.01)	482.71 (414.94)	0.03 (0.04)

* (-) indicates NA(NA); ** The values for the “unknown” category of the corresponding variables have been imputed for the modeling process; *** These participants were vaccinated on the day of blood sampling and thus considered as “not vaccinated”.

## Data Availability

Data are subject to data protection regulations and can be made available upon reasonable request to the corresponding author. To facilitate reproducibility and reuse, the code used to perform the analyses and generate the figures was made available in an open-source GitHub repository (https://gitlab.lrz.de/tropi-data-analysis-team/koco/kocoimpf-data.git (accessed on 11 July 2023)).

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
