# Peer review of "The Prospective COVID-19 Post-Immunization Serological Cohort in Munich (KoCo-Impf): Risk Factors and Determinants of Immune Response in Healthcare Workers"

_viruses, 2023, doi:10.3390/v15071574_

Round 1

Reviewer 1 Report

1.  These authors have reported information from a large study analyzing antibodies against COVID-19 in participants in Germany who have been infected or have been immunized or have both been infected and immunized.  This study included 6088 participants.  The anti-Spike protein seroprevalence was 94.7%.  The antinucleocapsid seroprevalence was 6.9%.  Healthcare worker status and contact with infected patients were definite risk factors for infection.  Vaccination and current smoking were associated with a reduced risk of infection.
2.  This study organization, the study size, the methods, and analysis provide very clear results.  The overall infection rate was relatively low.  The response to vaccination was high.
3.  The authors found that the antinuclear capsid immune responses were higher in older participants.  They suggest that this may be explained by more severe infections in older individuals with a higher production of antibodies.  However, the anti-Spike protein immune response after immunization was lower.  These results would suggest that natural infection is associated with an increased immune response in older individuals.  This analysis may be correct, but are there alternative possibilities?  Do they know anything about the symptom profile in individuals who had had prior COVID infections?  Has this been seen and other viral infections such as with influenza?  Is it possible that prior unrelated viral infections change immune responses to a new viral infection?  Also, their analysis of the effect of smoking seems somewhat counterintuitive?

Reviewer 2 Report

The manuscript by Reinkemeyer et al investigated the immune responses of health care workers in Munich in response to vaccination against and/or natural infection with SARS-CoV-2 during a period of 6 months in 2021. Several parameters were evaluated and several risk factors identified. 

The study is clear and well presented with a sound statistical analysis.

A weakness of the study is the relatively short study period and that it does not cover the circulation of the omicron VOC and therefore limits significantly the conclusions that can be drawn from the present study.

Some points I would like to be addressed:

In table 2 the number of unvaccinated persons with anti-N antibodies is significantly lower than those with anti-S (40 vs. 53) even though their antibody response can be caused only by a natural infections and therefore should theoretically always display both, anti-N and anti-S. Is it possible that the nucleocapsid assay is less sensitive? In that case it is likely that the number of natural infections is being underestimated by this approach.

The same picture can be seen in the same table in the section "time since infection", where the percentage of anti-N ranges from 11 to 70%, while theoretically 100% would be expected indicating a lack of sensitivity.

In the discussion section line 439 the authors refer to "the higher risk of infection among current smokers", where I believe a lower risk is correct.

In the discussion, newer study results covering the emergence of the omicron variant should be included  in order to put these results which cover infection by the Alpha and Delta variant into perspective.

English is acceptable, however, the manuscript would probably benefit from  editing by a native English speaker.
